# Yerba Maté (*Ilex paraguariensis*) Supplement Exerts Beneficial, Tissue-Specific Effects on Mitochondrial Efficiency and Redox Status in Healthy Adult Mice

**DOI:** 10.3390/nu15204454

**Published:** 2023-10-20

**Authors:** Chase M. Walton, Erin R. Saito, Cali E. Warren, John G. Larsen, Nicole P. Remund, Paul R. Reynolds, Jason M. Hansen, Benjamin T. Bikman

**Affiliations:** Department of Cell Biology and Physiology, Brigham Young University, Provo, UT 84602, USA

**Keywords:** yerba maté, metabolism, obesity, nutraceuticals, mitochondria

## Abstract

Yerba maté, a herbal tea derived from *Ilex paraguariensis*, has previously been reported to be protective against obesity-related and other cardiometabolic disorders. Using high-resolution respirometry and reverse-phase high-performance liquid chromatography, the effects of four weeks of yerba maté consumption on mitochondrial efficiency and cellular redox status in skeletal muscle, adipose, and liver, tissues highly relevant to whole-body metabolism, were explored in healthy adult mice. Yerba maté treatment increased the mitochondrial oxygen consumption in adipose but not in the other examined tissues. Yerba maté increased the ATP concentration in skeletal muscle and decreased the ATP concentration in adipose. Combined with the observed changes in oxygen consumption, these data yielded a significantly higher ATP:O_2_, a measure of mitochondrial efficiency, in muscle and a significantly lower ATP:O_2_ in adipose, which was consistent with yerba maté-induced weight loss. Yerba maté treatment also altered the hepatic glutathione (GSH)/glutathione disulfide (GSSG) redox potential to a more reduced redox state, suggesting the treatment’s potential protective effects against oxidative stress and for the preservation of cellular function. Together, these data indicate the beneficial, tissue-specific effects of yerba maté supplementation on mitochondrial bioenergetics and redox states in healthy mice that are protective against obesity.

## 1. Introduction

The unabated rise in cardiometabolic disorders such as heart disease, obesity, and type 2 diabetes mellitus necessitates the use of novel approaches to reduce associated risk factors. In recent years, scrutiny has been placed on dietary factors, with a particular emphasis on macronutrients (i.e., fats, proteins, and carbohydrates) [1] and meal frequency [2]. However, a growing body of evidence suggests that these strategies, while certainly relevant, may overlook the potential roles of other metabolites in mitigating cardiometabolic disease risk [3,4].

Secondary metabolites are naturally occurring substances produced by plants to make them more competitive in their environments, thus increasing their survival rate. These metabolites have historically been an important source of medicinal compounds and have been utilized in developing a number of therapeutics for diverse disease states [5]. Yerba maté is a herbal tea derived from the dried leaves of the South American *Ilex paraguariensis* plant and is widely consumed in South American countries such as Brazil, Argentina, Paraguay, and Uruguay. Yerba maté is a particularly rich source of secondary metabolites that have been implicated in improving cardiometabolic health. These compounds include purine alkaloids such as caffeine and theobromine, polyphenols such as phenolic acids and flavonoids, terpenes such as saponins and carotenoids, as well as numerous minerals and vitamins [5].

While some foods are known sources of secondary metabolites [6], environmental and farming trends have resulted in the steady reduction in these beneficial molecules in the foods we consume [7], increasing the need to obtain secondary metabolites from supplementary sources. Supplements enriched with these and other secondary metabolites have been shown to improve aspects of cardiovascular [3] and metabolic health [6,8], cognitive function [9], and more [10].

Indeed, yerba maté tea has been shown to favorably alter the lipid profile in humans [11]. Phenolic compounds such as chlorogenic acids, a class of phenol that is uniquely elevated in yerba maté tea [12], have been linked to several beneficial metabolic outcomes such as lipid homeostasis, weight management, cardiovascular protection, and glucose control [13]. Phenols and flavonoids have been shown to bolster antioxidant defenses via direct scavenging of free radicals [14] and to reduce inflammation by inhibiting the expression of proinflammatory cytokines [15,16]. For example, in response to high-fat diet-induced insulin resistance, a yerba maté treatment reduced inflammation by decreasing the nuclear translocation of NF-κB, a transcription factor that upregulates the expression of cytokines such as IL-6 and TNF-α [15]. These effects on antioxidant and anti-inflammatory defenses likely mediate some of the beneficial effects on metabolism and likely contribute to yerba maté’s protective properties in terms of aging.

Mitochondria are widely considered hubs for cellular metabolism because of their essential role in energy production. Impairments in mitochondria have been demonstrated in and have been suggested to drive metabolic disease [17]. Therefore, improving mitochondrial health has been proposed as a means of mitigating disease. In this study, we explore the effects of a four-week-long exposure to yerba maté on mitochondrial bioenergetics and redox potential within skeletal muscle, adipose, and hepatic tissue, demonstrating that yerba maté supplementation has beneficial metabolic effects that vary by tissue. Overall, this work adds to a host of evidence demonstrating that whole-body and tissue-specific metabolism are alterable via dietary intervention.

## 2. Materials and Methods

### 2.1. Animals

Six-month-old male and female C57BL/6 mice were group-housed at 22 ± 1 °C, 60–70% humidity, and were maintained on a 12 h light–dark cycle with ad libitum access to food. Mice were randomly divided into two groups and given free access to either water (control, CON) or yerba maté (MATÉ, Unicity International, Provo, UT, USA) for four weeks. Both groups of mice had ad libitum access to a standard rodent chow (LabDiet 5001, LabDiet, St. Louis, MO, USA) and their respective drinks. The yerba maté drink was prepared fresh daily as per the manufacturer’s instructions (7.49 g yerba maté powder per 10 oz water), and daily measurements of water and MATÉ consumption were recorded. The mice were weighed weekly. Equal numbers of males and females were used for each experiment.

Following conditioning, the mice were sacrificed, and tissues were collected (gastrocnemius skeletal muscle, subcutaneous adipose, and liver) and were either snap-frozen and stored at 80 °C for ATP quantification or kept on ice for mitochondrial respirometry and GSH/GSSG quantification.

Studies were conducted in accordance with the principles and procedures outlined in the National Institutes of Health Guide for the Care and Use of Laboratory Animals and were approved by the Institutional Animal Care and Use Committee at Brigham Young University (protocol #20-0203).

### 2.2. Tissue Permeabilization

Approximately 10–20 mg of each tissue was held on ice for mitochondrial respirometry. Samples were transferred to tubes containing 50 µg/mL of saponin in MiR05 respiration buffer [0.5 mM EGTA, 3 mM MgCl_2_, 60 mM K-lactobionate, 20 mM taurine, 10 mM KH_2_PO_4_, 20 mM HEPES, 110 mM sucrose, and g/L BSA (Sigma; A3803, St. Louis, MO, USA) adjusted to pH 7.1].

### 2.3. Mitochondrial Respirometry

High-resolution O_2_ consumption was determined at 37 °C in permeabilized muscle fiber bundles, adipose, and liver using the Oroboros O2K Oxygraph with MiR05 respiration buffer (Oroboros Instruments, Innsbruck, Austria), as described previously [18,19,20,21]. Before the addition of the sample into respiration chambers, a baseline respiration rate was determined. After the addition of the sample, the chambers were briefly hyperoxygenated to ~250 nmol/mL. Following hyperoxygenation, respiration was determined using all or parts of the following substrate–uncoupler–inhibitor–titration protocol: electron flow through complex I was supported using glutamate + malate (10 and 2 mM, respectively) to determine the leak oxygen consumption (GM). Following stabilization, ADP (2.5 mM) was added to determine the oxidative phosphorylation capacity (D). Succinate was added (S) for the complex I + II electron flow into the Q-junction.

### 2.4. ATP Quantification

ATP was quantified from frozen tissue samples with an ATPLite Luminescence Assay kit (Perkin Elmer). The frozen samples were thawed on ice and homogenized in mammalian cell lysis buffer provided by the manufacturer. The homogenates were diluted in water and were transferred to opaque 96-well plates in volumes of 100 µL per well. The ATPLite lysis buffer was added (50 µL) to each sample, and the plates were agitated for 5 min at 700 rpm at room temperature. The ATPLite substrate solution was then added (50 µL) to each well. The plates were covered with aluminum foil, agitated for an additional 5 min at 700 rpm at room temperature, and dark-adapted for 10 min. The luminescence was measured with a Victor Nivo Multimode Plate Reader (Perkin Elmer, Waltham, MA, USA).

### 2.5. Glutathione/Glutathione Disulfide Redox Potential Analysis

Methods for redox potential analysis have been previously described by Piorczynski et al. [22]. Isolated tissues were collected in 5% perchloric acid and boric acid (0.2 M) containing γ-glutamylglutamate (10 μM, Sigma-Aldrich, St. Louis, MO, USA) for GSH analysis. Concentrations of GSH and its oxidized form, glutathione disulfide (GSSG), were measured using reverse-phase high-performance liquid chromatography as S-carboxymethyl, *N*-dansyl derivatives. We then used the internal standard γ-glutamylglutamate for normalization, as described previously [23]. The proteins were acid-precipitated, and the samples were centrifuged at 16,000× *g* for 5 min; after, the soluble fraction containing free GSH and GSSG was derivatized. Samples were analyzed using an e2695 Separations Module (Waters, Milford, MA, USA) fitted with a Supelcosil LC-NH2 5 μm column (Sigma-Aldrich), and the detection of peaks was determined using a 2474 FLR Detector (excitation 335 nm and emission 518 nm, Waters). The GSH/GSSG redox potential (Eh) was calculated using the Nernst equation, on the basis of intracellular GSH and GSSG concentrations [24].

### 2.6. Statistics

Data are presented as the means ± SEM. The differences between the CON and MATÉ means were compared using Student’s *t*-test (GraphPad Prism 9; Microsoft Excel). Significance was determined as being at *p* < 0.05, a trend toward significance signaled by *p* < 0.15.

## 3. Results

### 3.1. Yerba Maté Consumption Prevents Weight Gain in Male and Female Mice

Over the four weeks of conditioning, female and male mice were maintained on a standard rodent diet. Cumulatively, the CON mice gained a significant amount of weight compared to the MATÉ mice (Figure 1A), which displayed no significant changes relative to their starting weight. Significant differences in the weights were observed between CON and MATÉ mice throughout the conditioning period, beginning at week 2 (Figure 1; week 1, *p* = 0.444; week 2, *p* = 0.0432; week 3, *p* = 0.0189; week 4, *p* = 0.0118). When assessed separately, both female (Figure 1B) and male (Figure 1C) mice displayed significant differences in weight over conditioning. Female MATÉ mice gained significantly less weight at week 4 (*p* = 0.0282), while male MATÉ mice gained significantly less weight at week 2 (*p* = 0.0368) and week 3 (*p* = 0.0172).

### 3.2. Yerba Maté Consumption Increases Mitochondrial Efficiency in Skeletal Muscle

High-resolution respirometry was performed on permeabilized gastrocnemius muscle fiber bundles and the results indicated a significant increase in the mitochondrial oxygen consumption between the CON and the MATÉ mice (Figure 2A) with the addition of ADP (*p* = 0.0296), but no significant differences with the addition of glutamate and malate (*p* = 0.216) or succinate (*p* = 0.481). The ATP quantification indicated a significant increase in ATP concentration in the homogenized gastrocnemius with yerba maté consumption (Figure 2B, *p* = 0.0023). The ratio of the gastrocnemius ATP concentration to oxygen consumption (ATP:O_2_), a measure of mitochondrial efficiency, was also significantly increased with yerba maté treatment compared to the control (Figure 2C, *p* = 0.0332).

To determine changes in the intracellular redox potential using the MATÉ treatment, concentrations of reduced and oxidized forms of glutathione, GSH and GSSG, respectively, were quantified. These analyses yielded a significant increase in GSH concentration (Figure 2D, *p* = 0.0491), but no significant difference in GSSG concentration (Figure 2E, *p* = 0.898) or the calculated E_h_ (Figure 2F, *p* = 0.2778).

### 3.3. Yerba Maté Consumption Decreases Mitochondrial Efficiency in White Adipose Tissue

The high-resolution respirometry performed on permeabilized adipose tissue collected from subcutaneous white adipose fat pads revealed significant increases in the mitochondrial oxygen consumption rates with yerba maté treatment (Figure 3A) with the addition of glutamate and malate (*p* = 0.0101), ADP (*p* = 0.0079), and succinate (*p* = 0.0367). The ATP quantification indicated a significant decrease in ATP production with yerba maté consumption compared to the use of water (Figure 3B, *p* = 0.0434). ATP:O_2_ ratios strongly trended toward a significant decrease (Figure 3C, *p* = 0.050007).

Redox analysis yielded no significant changes in GSH concentration (Figure 3D, *p* = 0.971), GSSG concentration (Figure 3E, *p* = 0.822), or GSH/GSSG E_h_ (Figure 3F, *p* = 0.993).

### 3.4. Yerba Maté Consumption Alters Hepatic Redox Potential but Not Mitochondrial Efficiency

The high-resolution respirometry performed on permeabilized liver tissue revealed no significant changes in the mitochondrial oxygen consumption rate (Figure 4A) between the control and the yerba maté-treated mice with the addition of glutamate and malate (*p* = 0.262), ADP (*p* = 0.297), or succinate (*p* = 0.882). No changes were observed in the ATP production (Figure 4B, *p* = 0.6141) or ATP:O_2_ ratio (Figure 4C, *p* = 0.315).

The redox analysis revealed a significant increase in GSH concentration (Figure 4D, *p* = 0.0449) but no significant change in GSSG concentration (Figure 4E, *p* = 0.334). This yielded a significant decrease in GSH/GSSG E_h_ (Figure 4F, *p* = 0.0251), indicating a shift toward a more reduced intracellular environment.

## 4. Discussion

Previous studies from our lab have demonstrated that diet has the capacity to alter mitochondrial efficiency in adipose [21] and brain tissue [25]. This study adds further support to the idea that diet alters mitochondrial bioenergetics. Specifically, these data demonstrate that yerba maté affects mitochondrial efficiency in a tissue-specific manner, such that maté consumption increases mitochondrial efficiency within skeletal muscle but decreases mitochondrial efficiency within adipose tissue. We interpreted both changes to be metabolically beneficial as the data agreed with our observation that yerba maté consumption protected mice against weight gain on a standard rodent diet (Figure 1). We also observed beneficial tissue-specific shifts in hepatic redox potential (Figure 4F), which further supports our positive interpretations.

While CON mice gained a significant amount of weight over the four weeks of conditioning, MATE mice experienced no significant changes compared to their baseline weight (Figure 1). Although we only observed a small, a ~1 g increase in body weight in the CON group and are uncertain whether this is physiologically relevant, this observation is consistent with other rodent and human studies that have demonstrated beneficial effects of yerba maté on lipid oxidation and weight management [26,27,28,29]. Indeed, multiple studies have demonstrated the anti-obesity effects of yerba maté in rodents that were fed obesogenic diets. For example, one study fed Swiss mice on a high-fat diet (HFD) for eight weeks [30]. These mice were then divided into two groups and fed with either water (vehicle control) or yerba maté for an additional eight weeks. The obese HFD-fed mice expectedly gained a significant amount of weight. This increase in weight was accompanied by systemic and hepatic insulin resistance and elevations in total cholesterol, high- and low-density lipoprotein cholesterol, and triglycerides. Yerba maté treatment attenuated, and in some instances completely reversed, these effects. In a hamster model of diet-induced hyperlipidemia, yerba maté consumption lowered the concentrations of serum lipids by modulating the expression of genes of lipid oxidation and increasing the activities of lipoprotein and hepatic lipases [27]. Another study in Wistar rats, maintained on a standard rodent diet supplemented with teas made from raw (not industrially processed) or commercial (industrially processed) maté, indicated no significant changes in body weight or plasma lipids over 30 days [31]. However, despite the lack of weight loss, the intra-abdominal fat pad mass was significantly reduced in the raw maté group, and the epididymal fat pad mass was significantly lower in both maté groups, indicating more efficient lipid oxidation. In our study, mice were also maintained on a standard rodent diet. However, we observed a significant increase in body weight in CON mice that was not observed in Silva et al.’s work [31]. Although we did not directly assess changes in the mass of intra-abdominal or epididymal fat pads, the differences in weight we observed are suggestive.

Anti-obesity effects of yerba maté supplements have also been observed in human clinical trials. For example, a double-blind, placebo-controlled study administered capsules of either yerba maté extract or a placebo to male and female subjects with obesity for 12 weeks [32]. In this time, they observed significant reductions in body fat mass and percentage, and in waist-to-hip ratios without any significant changes in safety parameters. These effects are supported by other trials [33]. Treatments using herbal preparations containing yerba maté, such as YGD (yerba maté, leaves of Ilex paraguayensis; Guarana, seeds of Paullinia cupana; and Damiana, leaves of Turnera diffusa), have also yielded reductions in body weight and body fat [28,29]. Andersen and Fogh observed a significant delay in gastric emptying after 45 days of YGD treatment that was taken before each main meal. They concluded this delay likely increased feelings of satiety and contributed to subsequent weight loss [29]. The study by Harrold et al. explored the effects of YGD and soluble fermentable fiber on appetite and energy intake by feeding participants an energy-controlled breakfast, administering treatments, and measuring appetite and energy consumption during lunch [28]. They observed significant reductions in both hunger and energy intake with YGD treatment that were further amplified with the addition of soluble fiber. While reductions in energy intake due to appetite suppression may be a contributing factor to the weight loss we observed in the mice in this study, the mitochondrial energy output was the focus of the work presented here and we assessed changes in metabolic rates of skeletal muscle, adipose, and hepatic tissue. However, it is also important to note that food intake and physical activity were not monitored in this study, which prevented us from attributing changes in energy input and output to specific processes beyond the mitochondria.

Here, we quantified changes in mitochondrial oxygen consumption, indicative of oxidative metabolism. We expected to observe differences in their metabolic responses to the MATÉ treatment because tissues inherently vary in their metabolic rates and functions. In skeletal muscle, no changes in oxygen consumption were observed with the addition of glutamate and malate, ADP, and succinate, which are substrates that feed into different complexes of the mitochondrial electron transport system (Figure 2A). However, significant increases in the ATP concentration (Figure 2B) and ATP:O_2_, a measure of mitochondrial efficiency, or ATP produced per unit of oxygen consumed (Figure 2C) were observed. This increase in efficiency indicates a more “coupled” metabolic state in which oxygen consumption is more tightly coupled to energy production. Interestingly, we observed somewhat of an opposite phenomenon in white adipose tissue. Adipose oxygen consumption rates increased with yerba maté treatment, indicating an increase in metabolic rate. While ATP production increased in skeletal muscle, maté administration induced a significant decrease in ATP production in white adipose tissue (Figure 3B). When combined with the changes in oxygen consumption, ATP:O_2_ ratios strongly trended toward a significant decrease (Figure 3C, *p* = 0.050007), or a more “uncoupled,” energetically inefficient state. All bioenergetic analyses performed in this study also accounted for sex, but no meaningful sex-specific effects were observed, as was the case with weight (Figure 1B,C).

Although the effect of yerba maté on exercise remains controversial [34,35], maté-induced changes in the mitochondrial bioenergetics in skeletal muscle and adipose have previously been associated with positive health outcomes, including improved low and high-intensity exercise performance [36] and increased lipid oxidation and fat loss [37]. Because yerba maté contains a number of bioavailable compounds, it is impossible to discern which compounds were directly responsible for these effects. However, weight loss observed with yerba maté consumption has been previously attributed to the caffeine content of the tea, which has been demonstrated to make up 1–2% of the dry leaf weight [38]. As a stimulant, caffeine has been shown to increase basal metabolic rate, improve aerobic performance and muscle oxygen content during exercise tests [39], and increase lipid oxidation [40,41]. Additionally, caffeine has been reported to increase the in vivo and in vitro expression of adipose UCP1 (uncoupling protein 1), a mitochondrial uncoupling protein that releases energy stored in the mitochondrial proton gradient in the form of heat in brown adipose [42]. Although we did not quantify changes in the white adipose UCP1 expression, the combination of our observations that yerba maté decreases white adipose mitochondrial efficiency and also induces weight loss, suggesting mitochondrial uncoupling and a loss of energy as heat. This has been shown to induce fat loss in humans and rodents and is consistent with studies which demonstrate that yerba maté stimulates mitochondrial biogenesis, thermogenesis, and white adipose “browning” [26]. This browning process makes white adipocytes phenotypically more similar to brown adipocytes, which are characterized by higher mitochondrial content and the expression of UCP1 [43].

Mitochondrial dysfunction and downstream oxidative stress are considered hallmarks of aging [44] and metabolic disorders [45,46]. Cellular homeostasis of oxidizing and reducing equivalents is normally maintained via enzymatic and nonenzymatic antioxidant systems that eliminate ROS and preserve cellular function. However, when the production of reactive oxygen species (ROS) exceeds cellular antioxidant defenses, redox states shift due to an increase in oxidizing equivalents and/or a decrease in reducing equivalents to promote a more oxidizing system. Oxidizing environments impaired cellular function [47] and promoted the oxidative modification of target macromolecules such as DNA, proteins, and lipids. Glutathione is the most abundant intracellular antioxidant thiol [48]. The GSH/GSSG couple is, therefore, commonly used as a proxy of cellular redox state and oxidative stress due to glutathione’s high concentration and involvement in first-line antioxidant defenses and as a direct free radical scavenger [47,49]. Here, we reported significant increases in GSH concentration in the skeletal muscle (Figure 2D) and the liver (Figure 4D) in response to yerba maté treatment, which may indicate a greater antioxidant buffering capacity. However, we only observed a significant change in GSH/GSSG E_h_ in the liver (Figure 4F). The significant decrease in GSH/GSSG E_h_ observed in the liver indicates a beneficial shift toward a more reduced environment, suggesting an enhanced cellular homeostatic function. Although we did not directly quantify ROS, the shift in GSH/GSSG is suggestive of decreased oxidative stress, which could be due to a reduction in the production of ROS, an upregulation of antioxidant defenses and free radical scavenging, or possibly both. This would require further investigation; however, our findings are consistent with other studies that report yerba maté attenuates oxidative stress [38,50].

The liver is a central player in whole-body energy metabolism due to its roles in digestion and carbohydrate, lipid, and amino acid metabolism. The liver also has essential functions in the metabolism and elimination of xenobiotics. As such, antioxidants are necessary for hepatic function. Indeed, the liver is the largest source of GSH, which not only participates in the detoxification of xenobiotics and determines redox status, but has also been shown to regulate other essential cellular functions such as cell proliferation and apoptosis [51]. Hepatic GSH production is impaired in liver diseases such as cholestatic liver injury, nonalcoholic fatty liver disease, and hepatocellular carcinoma, as well as metabolic disorders such as obesity, insulin resistance, type 2 diabetes, and cardiovascular diseases [52]. While oxidation of the GSH/GSSG E_h_ is associated with metabolic pathologies such as obesity, insulin resistance, type 2 diabetes, and cardiovascular disease, reduction in the GSH/GSSG E_h_ has been associated with hepatoprotection and a greater capacity to buffer oxidative stress. The favorable, pro-reducing shift in redox potential indicates an even greater, more enhanced protection against oxidative stress.

## 5. Conclusions

Although the literature on the effect of yerba maté on some aspects of exercise performance, such as muscle strength, remains controversial [34,35], there is a substantial amount of evidence indicating the beneficial effects of lipid oxidation and fat loss in rodents and humans [26,28,29]. Together, the data presented here demonstrate metabolically beneficial, tissue-specific effects of yerba maté on healthy mice such that maté increases energy efficiency in skeletal muscle but decreases efficiency in adipose, which prevents significant weight gain. Yerba maté has been repeatedly demonstrated to be safe in humans [32,33,52] and therefore can be said to represent a safe and accessible means of potentially improving metabolic function and protecting against obesity and other metabolic disorders.

## Figures and Tables

**Figure 1 nutrients-15-04454-f001:**
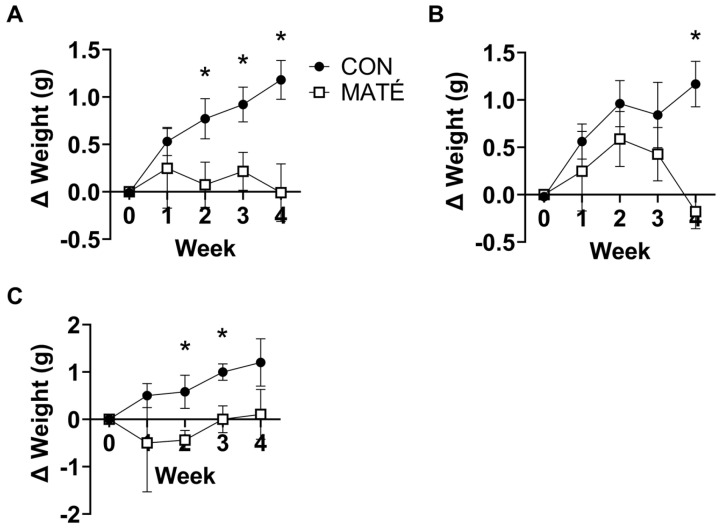
Yerba maté consumption prevents weigh gain in female and male mice. Body weight was measured weekly over the four weeks of yerba maté treatment and normalized to initial weights at week = 0. Female and male weights were assessed together (**A**), as well as separately (**B**,**C**). Female CON, *n* = 5; female MATÉ, *n* = 5; male CON, *n* = 5, male MATÉ, *n* = 5. * *p* < 0.05.

**Figure 2 nutrients-15-04454-f002:**
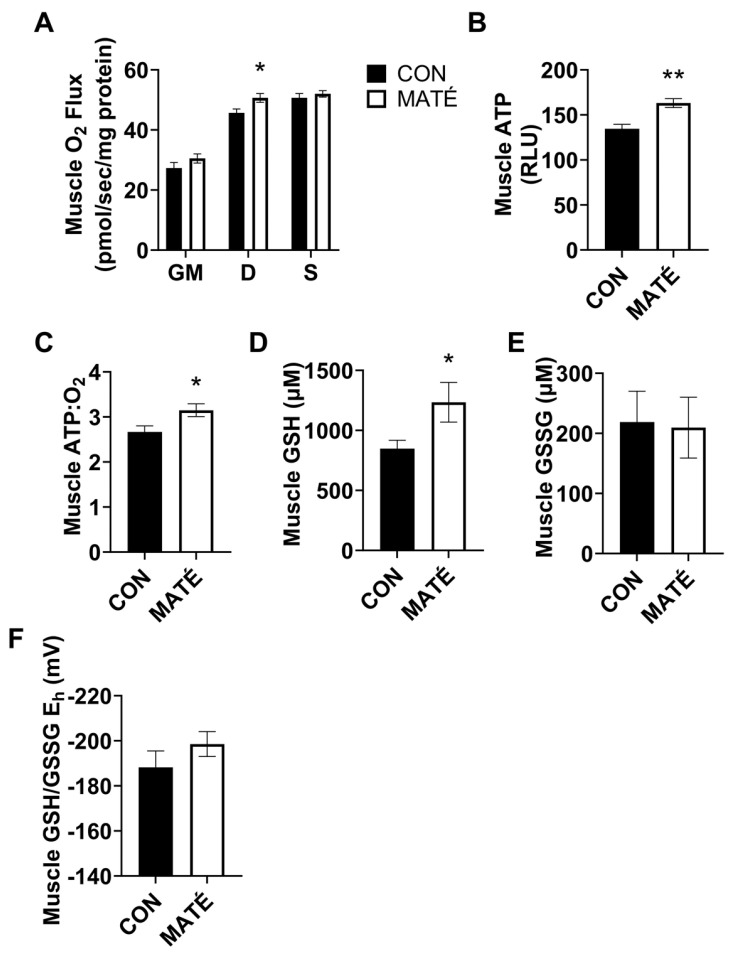
Yerba maté consumption increases the mitochondrial efficiency in skeletal muscle from male and female mice. Rates of mitochondrial oxygen consumption (**A**) were measured via a substrate–uncoupler–inhibitor–titration protocol from gastrocnemius tissue following four weeks of yerba maté supplementation. The protocol included the addition of glutamate and malate (GM), ADP (D), and succinate (S). ATP concentration was quantified (**B**) and ATP:O_2_ flux ratios (**C**) were calculated (CON, *n* = 6; MATÉ, *n* = 6). Concentrations of reduced glutathione (GSH, (**D**)) and oxidized glutathione (GSSG, (**E**)) were quantified via reverse-phase high-performance liquid chromatography. GSH/GSSG E_h_ redox potentials were calculated ((**F**); CON, *n* = 8; MATÉ, *n* = 8). * *p* < 0.05, ** *p* < 0.01.

**Figure 3 nutrients-15-04454-f003:**
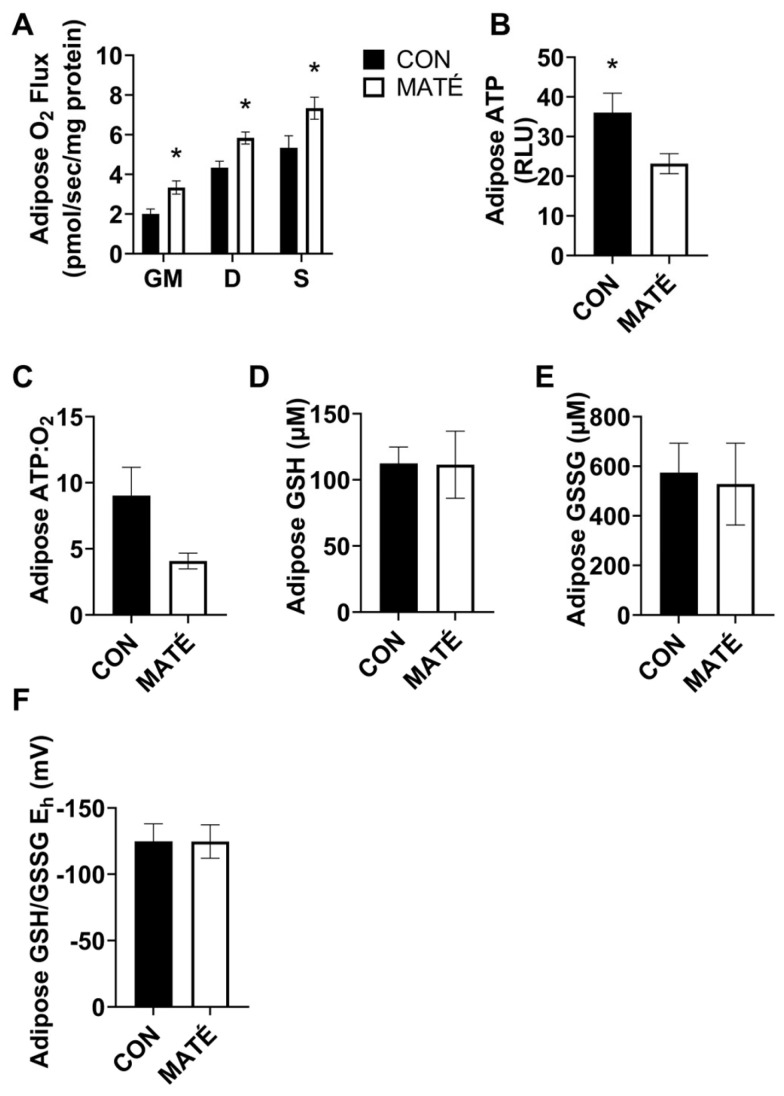
Yerba maté consumption decreases mitochondrial efficiency in white adipose tissue. Rates of mitochondrial oxygen consumption (**A**) were measured using a substrate–uncoupler–inhibitor–titration protocol from subcutaneous white adipose tissue following four weeks of yerba maté supplementation. The protocol included the addition of glutamate and malate (GM), ADP (D), and succinate (S). The ATP concentration was quantified (**B**), and ATP:O_2_ flux ratios (**C**) were calculated (CON, *n* = 6; MATÉ, *n* = 6). Concentrations of reduced glutathione (GSH, (**D**)) and oxidized glutathione (GSSG, (**E**)) were quantified via reverse-phase high-performance liquid chromatography. GSH/GSSG E_h_ redox potentials were calculated ((**F**); CON, *n* = 8; MATÉ, *n* = 8). * *p* < 0.05.

**Figure 4 nutrients-15-04454-f004:**
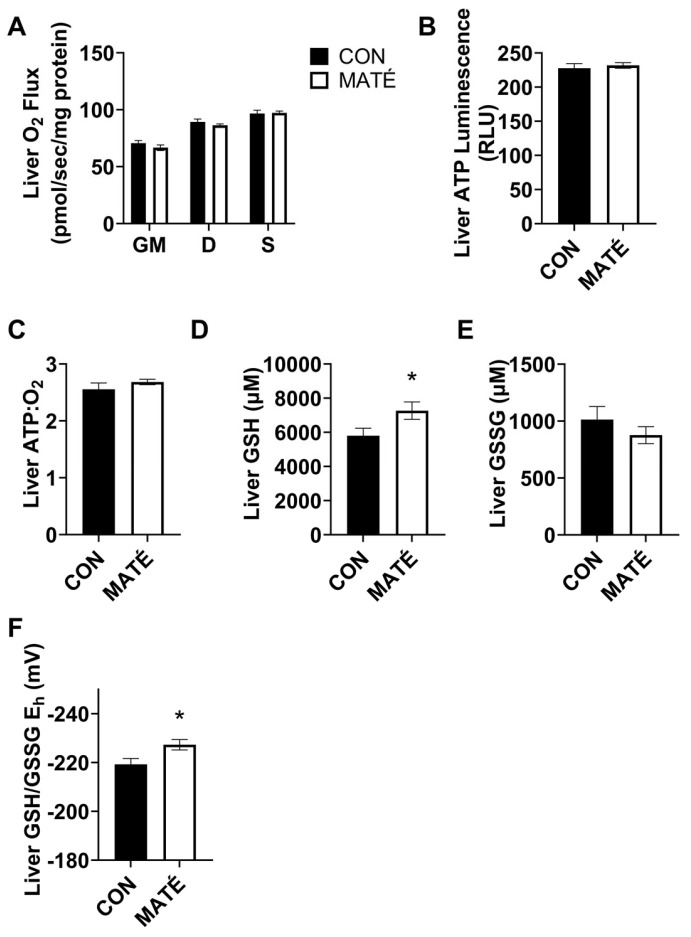
Yerba maté consumption alters hepatic redox potential but not mitochondrial efficiency. Rates of mitochondrial oxygen consumption (**A**) were measured via a substrate–uncoupler–inhibitor–titration protocol from liver tissue following four weeks of yerba maté supplementation. The protocol included the addition of glutamate and malate (GM), ADP (D), and succinate (S). The ATP concentration was quantified (**B**) and ATP:O_2_ flux ratios (**C**) were calculated (CON, *n* = 6; MATÉ, *n* = 6). Concentrations of reduced glutathione (GSH, (**D**)) and oxidized glutathione (GSSG, (**E**)) were quantified via reverse-phase high-performance liquid chromatography. GSH/GSSG E_h_ redox potentials were calculated (**F**); CON, *n* = 8; MATÉ, *n* = 8). * *p* < 0.05.

## Data Availability

Data can be provided by contacting the corresponding author.

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
