# Peer review of "Yerba Maté (Ilex paraguariensis) Supplement Exerts Beneficial, Tissue-Specific Effects on Mitochondrial Efficiency and Redox Status in Healthy Adult Mice"

_nutrients, 2023, doi:10.3390/nu15204454_

Round 1

Reviewer 1 Report

The article proposed by the authors is of great relevance since there are not many in vivo studies on the matter. In fact, there is one that they should add to the discussion: Dos Santos, T. W., Miranda, J., Teixeira, L., Aiastui, A., Matheu, A., Gambero, A., Portillo, M. P., & Ribeiro, M. L. (2018). Yerba Mate Stimulates Mitochondrial Biogenesis and Thermogenesis in High-Fat-Diet-Induced Obese Mice. Molecular nutrition & food research, 62(15), e1800142. https://doi.org/10.1002/mnfr.201800142.

The introduction is concise and adequate, although previous in vitro studies could be put into context in more detail.
The methodology describes that female and male mice are used, however, the number of mice used is not specified nor how many were male or female.
In fact, the results should be specified in separate graphs since there may be differences between males and females. You could specify the results separately and then together.
It should also specify how the yerba maté is prepared.
To have more consistent data, wertern blot or PCR studies should be performed. Icono de Validado por la comunidad

Icono de Validado por la comunidad         The text is well written with no spelling mistakes.

Reviewer 2 Report

The authors present the effects of yerba mate tea on mitochondrial efficiency and redox status. They present the differences for three tissues, muscle, liver and adipose tissue, which are the most relevant tissues in investigating metabolism and energy balance. While there is some novelty to the data, there are also major limitations or lack of crucial information.

1. The intervention is not adequately described. The authors state that the beverage was prepared fresh. However, there is no description of the material – where was the plant collected (in this case information on voucher specimen should be provided), which parts of the plant were used, where they dried or otherwise processed, did they use a commercially available product? How was the beverage prepared (amount of dry material per L, decoction/infusion, time of incubation?) What was the average daily intake of the drink (even is the drink was served ad libitum, the intakes should be given). Also, the beverage should be chemically characterized at least for the main bioactive components.

 2.      What was the number of mice and how was the sample size calculated? Figures 2-4 contain numerus, but not Figure 1. Where the groups equal in terms of sex?

 3.      Was the mitochondrial membrane permeability controlled and how?

 4.      The authors devote a large part of the discussion to the increase in body mass (or lack thereof). Although they briefly mention that energy output was the main focus, the interpretation of body mass should be interpreted with more caution. Important limitations, such as unmonitored food intake (if this was the case) and unmonitored physical activity, which prevent attribution of the difference to metabolic processes alone, should be stated. Do the authors believe that a 1g difference is physiologically relevant? This should be supported with citation.

 5.      Conclusion should be improved, a notion on muscle strength is irrelevant to the paper and the only statement referring to the data presented in this paper “…resulting in a significant weight loss” is incorrect. The same formulation should also be corrected in the abstract, as “weight loss” is not equal to “no weight gain”.

 6.      Figure captions contain several errors or lack of information; What is D in figure 2a, 3a? The FCCP is mentioned in figure caption, but its' use should be explained in the method section. All figure captions (Fig 2-4) state the material was from gastrocnemius tissue and not adipose and liver.

 7.      Minor: Line 192 should be mate, 269 – why capital letters, Line 92 should be O2

Round 2

Reviewer 1 Report

The authors have made the necessary changes and therefore it can be published.

Author Response

Thanks for your comments.